# Carbamoylated Erythropoietin Rescues Autism-Relevant Social Deficits in BALB/cJ Mice

**DOI:** 10.3390/neurosci6010025

**Published:** 2025-03-12

**Authors:** Amaya L. Street, Vedant P. Thakkar, Sean W. Lemke, Liza M. Schoenbeck, Kevin M. Schumacher, Monica Sathyanesan, Samuel S. Newton, Alexander D. Kloth

**Affiliations:** 1Department of Biology, Augustana University, Sioux Falls, SD 57197, USA; alstreet21@ole.augie.edu (A.L.S.); vpthakkar20@ole.augie.edu (V.P.T.); kmschumacher20@ole.augie.edu (K.M.S.); 2Division of Basic Biomedical Sciences, Sanford School of Medicine, University of South Dakota, Vermillion, SD 57069, USA; monica.sathyanesan@usd.edu (M.S.); samuel.sathyanesan@usd.edu (S.S.N.); 3Sioux Falls VA Healthcare System, Sioux Falls, SD 57105, USA

**Keywords:** autism, mouse model, erythropoietin, social behavior

## Abstract

Autism spectrum disorder (ASD) is a neurodevelopmental disorder that affects over 2% of the population worldwide and is characterized by repetitive behaviors, restricted areas of interest, deficits in social communication, and high levels of anxiety. Currently, there are no known effective treatments for the core features of ASD. The previous literature has established erythropoietin (EPO) as a promising antidepressant, working as a potent neurogenic and neurotrophic agent with hematopoietic side effects. Carbamoylated erythropoietin (CEPO), a chemically engineered non-hematopoietic derivative of EPO, appears to retain the neuroprotective factors of EPO without the hematologic properties. Recent evidence shows that CEPO corrects stress-related depressive behaviors in BALB/cJ (BALB) mice, which also have face validity as an ASD mouse model. We investigated whether CEPO can recover deficient social and anxiety-related behavioral deficits compared to C57BL/6J controls. After administering CEPO (40 μg/kg in phosphate-buffered saline) or vehicle over 21 days, we analyzed the mice’s performance in the three-chamber social approach, the open field, the elevated plus maze, and the Porsolt’s forced swim tasks. CEPO appeared to correct sociability in the three-chamber social approach task to C57 levels, increasing the amount of time the mice interacted with novel, social mice overall rather than altering the overall amount of exploratory activity in the maze. Consistent with this finding, there was no concomitant increase in the distance traveled in the open field, nor were there any alterations in the anxiety-related measures in the task. On the other hand, CEPO administration improved exploratory behavior in the elevated plus maze. This study marks the first demonstration of the benefits of a non-erythropoietic EPO derivative for social behavior in a mouse model of autism and merits further investigation into the mechanisms by which this action occurs.

## 1. Introduction

Autism spectrum disorder (ASD) is a neurodevelopmental disorder marked by socio-communicative deficits, repetitive and stereotyped behaviors, and restricted interests. The severity of each symptom lies along a spectrum, such that clinical presentation varies from patient to patient [1]. Patients with ASD show other comorbidities, like motor problems and anxiety, among others [1]. About 1 in 36 children born today are expected to receive an ASD diagnosis [2], and an ASD diagnosis can impose a potentially heavy economic burden, estimated to be USD 3.6 million [3]. Because of the significant public health challenges posed by ASD, decades of preclinical research have sought to uncover effective therapeutic strategies [4,5,6]. Unfortunately, this work has, so far, yielded few effective therapies, with the only existing pharmacological therapies treating symptoms, rather than core ASD traits [7].

Still, this body of work has revealed features of ASD pathophysiology that merit further consideration. Studies in both ASD patients and mouse models of ASD have demonstrated disrupted learning and memory [8]; dysregulation of synaptic function and plasticity [9,10,11]; altered neurotrophic function [12,13]; disrupted neurogenesis [14,15]; and increased neuroinflammation [16,17]. These features may yet be fruitful targets for drug development.

Erythropoietin (EPO) may be an intriguing therapeutic possibility for addressing some or all of these aspects of ASD pathophysiology. Over the last two decades, this endogenous cytokine and anemia treatment has emerged as a potential treatment for a variety of neurological and psychiatric disorders, most notably depression [18,19]. Studies into EPO’s role in the brain have revealed that it is involved in memory improvement; improved synaptic plasticity; increased neurotrophic signaling and neurogenesis; and reduced neuroinflammation [19,20,21,22]. Because of the remarkable overlap between EPO effects and ASD pathophysiology, it is reasonable to hypothesize that EPO could rescue the clinical features of ASD, though few studies have investigated this hypothesis specifically.

However, a critical limitation is that EPO at clinically relevant doses has adverse side effects due to its hematopoietic activity [18]. In response, the field has shifted its focus to developing EPO analogs that retain the neurological effects but preclude their hematological effects. One such drug is carbamoylated EPO (CEPO), prepared by carbamoylating key lysine residues of EPO [23,24]. Like EPO, CEPO improves mood and cognition in rodents [21,24,25,26], particularly in models of depression, like the stress-susceptible inbred mouse BALB/c [26]. These CEPO effects appear to correspond with enhancements in synaptic plasticity, neurotrophic signaling, neuroinflammation, and neurogenesis [21,25,26,27,28]. Importantly, CEPO is safe and effective for extended use: ten doses of CEPO at 30 μg/kg yielded no increase in hematocrit in BALB/c mice [25], and CEPO delivered at a non-erythropoietic dose of 40 μg/kg [23] twice weekly for six weeks yielded robust cognitive improvements [21]. Thus, CEPO appears to be a suitable alternative for testing the efficacy of non-erythropoietic EPO-like molecules in ASD.

In the present study, we leveraged the BALB/cJ (BALB) mouse to determine whether CEPO can rescue behavioral phenotypes related to ASD. In addition to its depression-related behavioral phenotypes, the BALB/c mouse shows low sociability—a key phenotype for ASD in model mice—in the three-chamber social approach task compared to the more social C57BL/6J (C57) mice. On the basis of this sociability deficit and other behavior and anatomical and synaptic phenotypes that parallel what has been seen in some human ASD patients, it has been suggested that BALB mice can model some aspects of ASD preclinically [29]. We found that chronic delivery of CEPO rescued sociability—a key phenotype for ASD in model mice—while also alleviating other anxiety-related behaviors relevant to ASD, including in the elevated plus maze. Overall, this study establishes CEPO as a promising therapy for treating several core traits of ASD.

## 2. Methods and Materials

### 2.1. Animals

Male BALB/cJ (BALB) mice and male C57BL/6J (C57) mice were obtained from Jackson Laboratories (Bar Harbor, ME, USA; BALB/cJ, #000651, C57BL/6J, #000664).

The animals arrived at 7 weeks of age and were habituated in the vivarium for 1 week. Drug or vehicle administration began at 8 weeks of age. In total, 45 mice were used in this experiment, 24 BALB mice and 21 C57 mice (Table 1). The mice were housed with a 12 h light/dark cycle in open-top mouse cages (Ancare, Bellmore, NY, USA), with 3–4 mice in each cage. All mice were housed with other animals from the same experimental group. Cages were cleaned twice a week, and all mice had ad libitum access to food and water. All experimental procedures complied with the regulations set by the Augustana University Institutional Animal Care and Use Committee.

### 2.2. Drug Administration

Carbamoylated erythropoietin (CEPO) was generated at the University of South Dakota according to previously published procedures [24,26,28,30]. CEPO dissolved in sterile phosphate-buffered saline (PBS, pH 7.4; vehicle) was administered intraperitoneally every other day for 21 days after the mice reached eight weeks of age. The CEPO concentration used was 40 μg/kg, which did not induce hematological changes compared to the vehicle, as observed in previous studies with erythropoietin and non-hematopoietic erythropoietin derivatives [21,28]. Of the 24 BALB mice, 12 received CEPO, and 12 received the vehicle only. Of the 21 C57 mice, 11 received CEPO, and 10 received the vehicle only. The mice were weighed before every drug administration to ensure the appropriate doses were delivered and to monitor their health.

### 2.3. Behavioral Timeline

Following the drug administration period, all mice were subjected to four behavioral tasks (Figure 1). These tasks were selected to evaluate the effects of CEPO on social behaviors (three-chamber social approach task), anxiety behaviors (open field and elevated plus maze), and depression-like behaviors that had been previously evaluated in BALB mice [26]. All behavioral experiments took place in a dedicated room under low light (approximately 4 lux) conditions in the presence of white noise (approximately 50 dB; Noise & Co Colored Noise Generator, https://mynoise.net/NoiseMachines/whiteNoiseGenerator.php, accessed on 5 March 2025).

### 2.4. Three-Chamber Social Approach Task

The three-chamber social approach task aided in the quantification of the sociability of the mice according to previously published protocols [31]. Before the task, the mice were allowed to acclimate in the behavioral room for 30 min. This task consisted of three phases, with each phase lasting 10 min. In the first phase, the mouse explored a custom-built three-chamber maze [32] without any novel objects or other mice present. In the second phase, two empty wire mesh pencil cups (Amazon Basics) were placed upside down, one in each of the outermost chambers, and the mouse was again free to explore the maze. Finally, in the third phase, a strain-matched, aged-matched novel stranger mouse was placed in one of the empty cups, and a novel non-social object (plain wooden block) was placed in the opposite cup. Again, the mouse was free to explore the entire space. Between the experimental phases, the mouse was temporarily placed in a separate open-top cage. After the task, the three-chamber maze was cleaned using a 70% ethanol solution.

Each phase was recorded using a PSEye camera (Sony Interactive Entertainment, San Mateo, CA, USA) mounted on the ceiling [33]. Each video was analyzed by two individuals blinded to the compound injected into the mouse. The phase two recordings were timed manually and analyzed to determine whether there was any side bias that might be present in any individual mouse; to do so, the durations of the interactions were measured, defined as directed exploratory behavior within 3 cm of an object, with the cups on either side of the maze. Side bias was calculated according to the equation [(left chamber time − right chamber time)/(left chamber time + right chamber time)]. Mice were excluded if the side bias was higher than ±0.7.

The phase three recordings were timed manually and analyzed to determine social and non-social behaviors in the presence of the novel stranger mouse. The following parameters were collected: time spent interacting with both of the cups, defined as directed exploratory behavior within 3 cm of an object; the number of 5 s interaction bouts with each object; the amount of time spent grooming; the number of 5 s bouts of grooming; and the number of rears [31]. Climbing the cup without interest in the object/animal inside was not counted as social behavior. Rears were defined as vertical movement on the hind legs, either using a wall for support or in the open. Finally, to determine sociability, the sociability index was calculated as [(interaction time with social object − interaction time with non-social object)/(interaction time with social object + interaction time with non-social object)], as per previous studies [31]. A positive sociability index denotes increased interaction time with the novel stranger mouse.

### 2.5. Open Field

The open field task is one way to measure the anxiety and exploratory activity of mice [34]. The task was carried out as previously described [33]. Prior to the task, the mice were allowed to acclimate in the behavior room for 30 min. The open field maze consisted of a large white Plexiglas box with dimensions of 41 cm × 41 cm × 41 cm and an open top. The mice were allowed to explore the maze for 20 min. The mice were recorded using a PSEye camera (Sony Interactive Entertainment, San Mateo, CA, USA) mounted on the ceiling. The distance traveled and percentage of time spent in the center of the maze were calculated using a custom MATLAB R2024b (RRID: SCR_001622) code, as in previous studies [33]. The center was defined as 10.25 cm from the walls. Rears were counted and defined as standing on the hind legs, either using a wall for support or in the open. The time spent grooming, as well as the number of 5 s bouts of grooming, was also recorded.

### 2.6. Elevated Plus Maze

The elevated plus maze acted as a second measure of anxiety [35]. The elevated plus maze was 50 cm off the ground: all arms were 50 cm long when measured from the center and 10 cm wide. Two arms on opposite sides of the maze had walls that were 30 cm high; the other two arms were open. Before the task, the mice were allowed to acclimate in the behavior room for 30 min. To start the task, the mice were placed in the center of the plus and were allowed to explore the maze for 10 min. While in the maze, the mice were recorded using a PSEye camera (Sony Interactive Entertainment, San Mateo, CA, USA) mounted on the ceiling. The measurements were performed manually, and the metrics analyzed included the number of rears, the number of times the mice entered the open arms, and the time spent grooming.

### 2.7. Forced Swim

The forced swim task measures the tenacity of the mice under stress [36]. The experiment was carried out according to previously published protocols [33]. Preceding the task, the mice were acclimated to the behavior room for 30 min. A 3 L beaker was filled ¾ of the way with water at approximately 25 °C. The mice were placed in the water for a total of 6 min. For the first 2 min, the mice acclimated to the water; the final 4 min were recorded and analyzed. To analyze this task, the amount of time the mice spent actively paddling (using hind and front limbs to produce rhythmic and propulsive movement) was compared to the amount of time spent immobile.

### 2.8. Statistics

All statistical analyses were carried out in GraphPad Prism 9 (GraphPad Software, San Diego, CA, USA; RRID:SCR_002798) using the two-way analysis of variance (ANOVA) test with post hoc comparisons. Post hoc comparisons among the groups were carried out following the discovery of statistically significant effects among the groups. The Bonferroni-corrected planned comparisons were computed between C57-vehicle mice and BALB-vehicle mice, BALB-vehicle mice and BALB-CEPO mice, and BALB-CEPO mice and C57-vehicle mice (Figure 2, Figure 3(A,D1,D2), and Figure 4) or between social object-vehicle and social object-CEPO groups, social object-vehicle and non-social object-vehicle groups, social object-CEPO and non-social object-CEPO groups, and non-social object-vehicle and non-social object-CEPO groups (Figure 3(B1–C2)). The homoscedasticity of residuals was confirmed using Spearman’s rank correlation test, and the normality of residuals was confirmed using the Shapiro–Wilk test. In some cases, the data were transformed to satisfy the assumptions of two-way ANOVA using $\sqrt{y+0.01}$ or $\sqrt{y-1}$ transformations. These cases are indicated in Appendix A. A *p*-value less than 0.05 is considered statistically significant unless otherwise specified. A complete report of statistical comparisons in this study can be found in Appendix A.

## 3. Results

Following a series of 11 injections every other day, the mice underwent behavioral testing (Figure 1). The complete results from the statistical analyses can be found in Appendix A.

### 3.1. Weights

Unlike erythropoietin administration, CEPO injections have been shown to have a negligible effect on the overall health of mice [31,35]. To monitor the effects of CEPO on the physical health of the mice, each mouse was weighed before the injection period and before each subsequent dose. Here, we report on the changes in weights from the first injection to the eleventh injection. The percentage weight changes for the C57 and BALB mice were not significantly different between the treatment groups after the complete course of injections, regardless of mouse strain (Figure 2; two-way ANOVA, treatment effect: *p* = 0.6759). However, there was a small but noticeable strain difference (Figure 2; two-way ANOVA, strain effect: *p* = 0.0979).

### 3.2. Three-Chamber Social Approach Task

We used the three-chamber task to examine whether CEPO could revert well-documented social-approach deficits in male BALB mice to C57 levels. There was a considerable difference in the sociability between the strains (Figure 3A; two-way ANOVA with planned comparison, strain effect: *p* = 0.0008), and the sociability deficit in the BALB mice (vehicle mice compared to C57 vehicle mice: *p* = 0.0015) was rescued by CEPO (Figure 3A; two-way ANOVA with planned comparisons, treatment effect: *p* = 0.0388; interaction: *p* = 0.0456; BALB vehicle vs. BALB CEPO: *p* = 0.0298; C57 vehicle vs. BALB CEPO: *p* = 0.7453). Analyzing the interaction time and number of interactions more closely revealed that the improvement in the sociability index was driven almost entirely by an increase in the interaction time with the social object (Figure 3(B1); two-way ANOVA with planned comparisons, object effect: *p* = 0.0323; treatment effect: *p* = 0.0130; interaction: *p* = 0.27; social object vehicle vs. social object CEPO: *p* = 0.0493). There was a modest but statistically insignificant difference between the social object interaction time and the non-social object interaction time (*p* = 0.0803), likely contributing to the difference in sociability. There was no similar significant difference in the amount of time spent exploring the non-social object (non-social object vehicle vs. non-social object CEPO: *p* > 0.9999), and the BALB vehicle mice showed no significant preference for the social object over the non-social object (*p* > 0.9999). There was no evidence of a concomitant difference in the number of interactions with either object between the vehicle and CEPO BALBs (Figure 3(B2); two-way ANOVA, treatment effect: *p* = 0.1173; object effect: *p* = 0.1119; interaction: *p* = 0.2257). For the C57 mice, no significant difference was observed between the CEPO and vehicle groups (Figure 3(C1); two-way ANOVA, treatment effect: *p* = 0.8962; interaction: *p* = 0.2257). As expected, there was a significant difference in the time spent with the social vs. non-social object (Figure 3(C1); two-way ANOVA, object effect: *p* < 0.0001). These differences between the vehicle and CEPO C57 groups (social object-vehicle vs. non-social object-vehicle: *p* < 0.0001; social object-CEPO vs. non-social object-vehicle: *p* < 0.0001) likely account for the positive sociability in both groups. Likewise, for the number of interactions, treatment yielded no significant impact (Figure 3(C2); two-way ANOVA with planned comparisons, treatment effect: *p* = 0.7142; interaction: *p* = 0.5679), but there was a significant object effect (*p* < 0.0001). There was no significant difference in the time spent grooming between treatments, but there was between the strains of mice (Figure 3(D1); two-way ANOVA, treatment effect: *p* = 0.4675; strain effect: *p* = 0.0269; interaction: *p* = 0.4707). However, no planned comparison showed significant differences between the groups (*p* > 0.05). There was no significant difference in the number of rears due to strain or treatment (Figure 3(D2); two-way ANOVA, treatment effect: *p* = 0.4675; strain effect: *p* = 0.1670; interaction: *p* = 0.5784).

### 3.3. Open Field Task

We used the open field task to determine whether CEPO could rescue anxiety-related exploratory behaviors in the BALB mice relative to the C57 controls [37]. While we were able to recapitulate the anxiety-related differences in rears (Figure 4A; two-way ANOVA with planned comparisons, strain effect: *p* < 0.0001; BALB vehicle vs. C57 vehicle: *p* < 0.0001) and total distance traveled (Figure 4B; two-way ANOVA with planned comparisons, strain effect: *p* < 0.0001; BALB vehicle vs. C57 vehicle: *p* = 0.0713) between the BALB and C57 mice, there were no significant differences between the treatments in the C57 or BALB mice for rears (Figure 4A, treatment effect: *p* = 0.9100; interaction: *p* = 0.9728) or total distance (Figure 4B, treatment effect: *p* = 0.2770; interaction: *p* = 0.4237). Being generally more anxious, the BALB mice exhibited less exploratory behaviors than their C57 counterparts, independent of receiving the CEPO treatment. We were also able to recapitulate a strain difference in the time spent in the center of the open field (Figure 4C; two-way ANOVA with planned comparisons, strain effect: *p* < 0.0001) and observed an effect on treatment (treatment effect: *p* = 0.0282) with no interaction (*p* = 0.1425). However, this effect was modest at best, as it was not accompanied by statistically significant improvements in either group (C57 CEPO vs. C57 vehicle: *p* = 0.2338; BALB CEPO vs. BALB vehicle: *p* = 0.279).

### 3.4. Elevated Plus Maze

We used the elevated plus maze as a second measurement of anxiety in the BALB mice, as it has been observed to produce different results compared to C57 mice [38]. We confirmed a difference between the strains (Figure 4D; two-way ANOVA with planned comparisons, strain effect: *p* = 0.0212), with a statistical difference between the vehicle groups (C57 vehicle vs. BALB vehicle: *p* = 0.0463). In the BALB mice, treatment with CEPO significantly increased the number of entries the mice made into the open arms (treatment effect: *p* = 0.0212; BALB interaction: *p* = 0.1425; BALB vehicle vs. BALB CEPO: *p* = 0.0295). This increase constituted a rescue in the number of entries made into open arms, as the BALB-CEPO levels were statistically indistinguishable from the C57-vehicle levels (*p* > 0.9999). CEPO had no effect on C57 behavior (*p* = 0.9538).

### 3.5. Porsolt’s Forced Swim Task

Finally, we used the forced swim task to compare our experiments with prior experiments using CEPO in BALB mice. We observed a modest decrease in the immobility time due to treatment, which did not reach a level of statistical significance (Figure 4E; two-way ANOVA with planned comparisons, treatment effect: *p* = 0.0516). Moreover, we did not observe an appreciable difference in immobility between the strains (strain effect: *p* = 0.1983; interaction: *p* = 0.6062).

## 4. Discussion

We hypothesized that carbamoylated erythropoietin (CEPO) could rescue autism spectrum disorder (ASD)-related behaviors that appear in the BALB/cJ (BALB) mouse model. Like erythropoietin (EPO), CEPO may have a pharmacological profile that may alter the underlying pathophysiology of ASD and thus affect the behaviors that characterize the disorder. Using a battery of tasks that demonstrate significant behavioral differences between BALB and C57 mice [37,39], we were able to rescue deficits in the three-chamber social approach task and the elevated plus maze task, but not in the open field or forced swim tasks. These findings suggest that the administration of CEPO may have value for treating ASD, but the generalizability of these results to the wide variety of mouse models merits further consideration.

We were most interested in whether CEPO could rescue disrupted social behavior, the primary ASD-relevant phenotype in the BALB mouse [29,40]. Consistent with other studies examining social approach in BALB mice, we found that they had significantly reduced sociability compared to the age-matched C57 controls, particularly in settings with restricted social contact, like the three-chamber maze [37,38,41,42,43,44]. The present study is notable in that it found a more pronounced sociability difference in adult male mice, whereas other studies have suggested that the sociability deficit seen in juvenile [37,44] and adolescent [38,43] BALB mice improves with age, as compared to C57 mice. Our observations may be explained by our use of a standardized, three-phase protocol for the three-chamber task, which has been demonstrated to detect social deficits more sensitively [31]. Semi-chronic administration of CEPO was able to rescue this deficit in the BALB mice. This improvement was largely driven by an increase in the amount of time they interacted with the stranger mouse, with no change in time spent interacting with the non-social object or in other exploratory behaviors. This finding indicates that CEPO decreases putative social interaction deficits in BALB mice [45], without altering the overall activity in the task.

Our study is the first demonstration of a role for non-hematopoietic EPO derivatives in rescuing social dysfunction, and it is only the fourth study to examine EPO or its derivatives in an ASD rodent model [46,47,48]. The three prior studies looked at social behavior in environmentally induced rat models of autism. Solmaz and colleagues (2020) found that daily EPO administration of a relatively high dose improved social interaction time in liposaccharide-exposed male rats, but their sociability overall was not addressed. Hosny and colleagues (2023) made a similar observation in EPO-treated propionic acid-exposed rats, albeit in an open field test. Haratizadeh and colleagues (2023) observed that early postnatal EPO administration did not rescue sociability in valproic acid-exposed rats but did improve their social preference, indicating an improvement in memory, which is consistent with other studies. Other studies investigating the therapeutic potential of erythropoietin have demonstrated its ability to rescue three-chamber social approach behavior in rodent models of other neurodevelopmental disorders, including prenatal brain injury [49]. Together, this evidence suggests a role for EPO in modulating social behavior. Our study builds on this work by exploring the role of an EPO derivative using the “gold standard” of social behavior—the three-chamber social approach task—in an inbred mouse that has some face validity for ASD [29]. Future work could leverage transgenic mouse models of ASD-related single-gene disorders to understand the generalizability of the effects of CEPO on social behavior and uncover the mechanisms underlying these actions.

The BALB mouse also demonstrates anxiety-related behavior, an ASD-related behavior that might also explain the social deficits [38,42]. In our study, we documented that BALB mice exhibited significantly less time in the center of the open field and significantly fewer entries into the open arms of the elevated plus maze, compared to the C57 controls. This is consistent with other studies [38]. However, where CEPO was able to rescue the number of open-arm entries, it made a small but statistically insignificant improvement in the mice’s behavior in the open field. This finding in the open field is consistent with previous studies on the acute effects of CEPO in BALB mice [26]. Our results suggest that CEPO improves a mechanism underlying one distinct aspect of anxiety-related behavior, but not others; indeed, the findings from the elevated plus maze and the open field in BALB mice and other strains are often weakly correlated [50]. Moreover, it remains to be seen whether this effect in the elevated plus maze is related to the change in social behavior. A small number of studies have shown a weak correlation between elevated plus behavior and social behavior in BALB mice [38,45], and none of the previous studies on the effects of EPO and its derivatives in social behavior also examined anxiety-related behavior. It is entirely possible that CEPO acts on distinct mechanisms responsible for social and anxiety-related behaviors.

It should be noted that we were unable to replicate the rescue of depression-like behavior in the forced swim that had been previously observed with CEPO in BALB mice [26], and with EPO in some studies with other rodent models [51]. We speculate that these differences might be accounted for by slight differences in the forced-swim protocol used in our laboratory, or due to washout several days after the last CEPO injection. Prior studies have evaluated the effects of EPO and its derivatives delivered just hours before the forced-swim task [21,26,51]. This point highlights the importance of evaluating the long-lasting impacts of CEPO after cessation of the drug, not just for depression-like behaviors but for ASD-relevant behaviors as well.

The mechanism by which CEPO affects social behavior in BALB mice remains unknown. CEPO has been demonstrated to elevate levels of neurotrophic factors, including brain-derived neurotrophic factor (BDNF) [24,28]; engage immediate early genes and other mechanisms that modulate synaptic function [21,28,30]; regulate neuronal development and dendritic outgrowth [25,27,28]; alter the neuroinflammatory response and modulate glial function [25,52]; and regulate neurogenesis [21], among other functions [18]. Although these findings were primarily restricted to the hippocampus—which plays a role in social memory—it is possible that CEPO causes similar alterations in other parts of the social circuitry. However, it is unclear which mechanisms would be responsible for our observations in this study. The few studies that have addressed the mechanisms underlying social dysfunction in BALB mice have focused on improving glutamatergic transmission [53,54]—suggesting overlap with the pro-neuroplastic properties of CEPO—or on altering the serotonin–oxytocin pathway [55]. These studies do not exclude the possibility that other mechanisms are involved. In this direction, there is evidence that the mechanisms engaged by CEPO, EPO, and other EPO derivatives might correct social behavior in ASD animal models. For example, in their study of EPO administration on liposaccharide-exposed rats, Solmaz et al. observed diminished TNF-ɑ levels and reduced glial activity concomitant with improvements in social behavior [48]. Another study observed a similar relationship between glial activity and social behavior using an EPO derivative [56]. Still, other studies that have used non-EPO compounds that engage similar mechanisms have shown improvements in sociability [57,58,59,60,61]. Future work should examine the mechanisms underlying social dysfunction in BALB mice and whether CEPO directly influences those mechanisms in a manner that correlates with social improvement.

There are several ways that our study—focused on delivering CEPO to a single mouse line demonstrating ASD-relevant features, semi-chronically in young adulthood—should be extended to understand how generally effective CEPO and other EPO derivatives can be as ASD treatments. The BALB mouse is by no means representative of all ASD cases. For instance, one study that used neuroimaging to understand the heterogeneity of ASD mouse models showed that BALB mice are included in one of three clusters, suggesting important pathophysiological differences among models of the same disorder [62]. In addition, other mouse models—while also demonstrating more phenotypes relevant to ASD—have construct and predictive validity that has not been demonstrated in BALB mice, and are thus more relevant and robust preclinical tools [63]. It will be necessary to extend our work to other ASD mouse models outside of the BALB cluster to determine whether CEPO or other EPO derivatives are generally effective at rescuing social disruption, and whether that rescue is associated with similar changes at the cellular and molecular levels. It will also be necessary to understand how sex plays a role as a biological variable—both from the standpoint of sex differences in social dysfunction and in sex differences in CEPO effectiveness. In addition, the limits of CEPO administration remain to be seen. It will be important to examine whether CEPO administration is more or less effective at different developmental time points, as ASD is a chronic disorder [64]. Furthermore, it is crucial to understand how long the effects of semi-chronic dosing of CEPO last. Answering these questions will be essential for understanding how to extend this limited study to a more broadly effective eventual treatment for ASD patients.

## 5. Conclusions

We demonstrate for the first time that an erythropoietin-derived compound can address social features of autism spectrum disorder in mice. Further work will be needed to generalize and extend this work to other mouse models and a broader suite of autism-related phenotypes, and to understand the mechanisms by which erythropoietin-like compounds act to alter these phenotypes.

## Figures and Tables

**Figure 1 neurosci-06-00025-f001:**
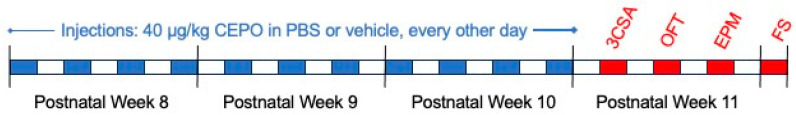
Experimental timeline for this experiment. Drug administration (blue) took place over three weeks, starting 8 weeks postnatal, with a period of behavioral testing (red) following drug administration. Each division corresponds to one day. PBS, phosphate buffered saline; 3CSA, three-chamber social approach task; OFT, open field task; EPM, elevated plus maze task; FS, Porsolt’s forced swim test.

**Figure 2 neurosci-06-00025-f002:**
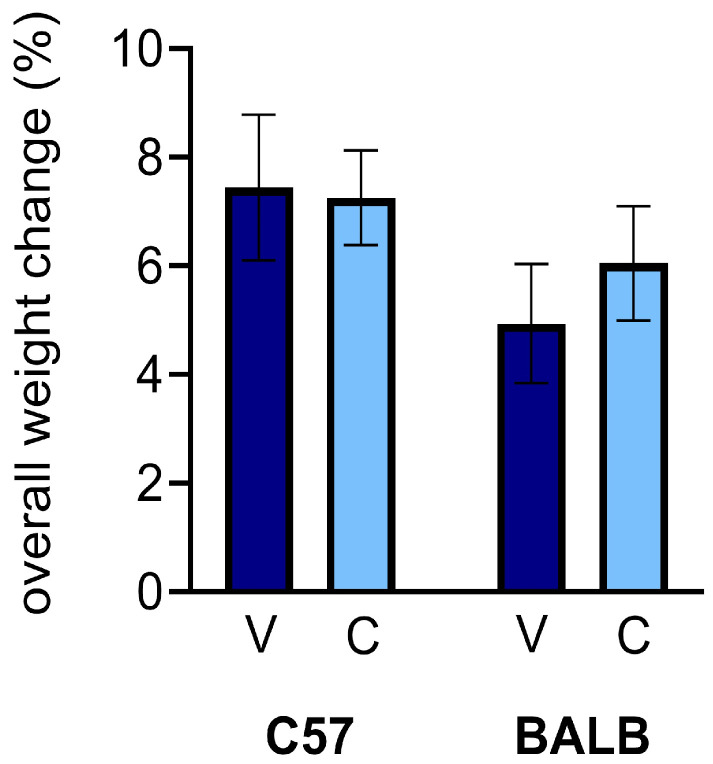
CEPO administration did not significantly impact the weights of mice of either strain compared to vehicle injection (two-way ANOVA followed by planned comparisons, *p* > 0.05). Dark blue, vehicle (V) injections; light blue, CEPO (C) injections. Error bars represent ± standard error of the mean.

**Figure 3 neurosci-06-00025-f003:**
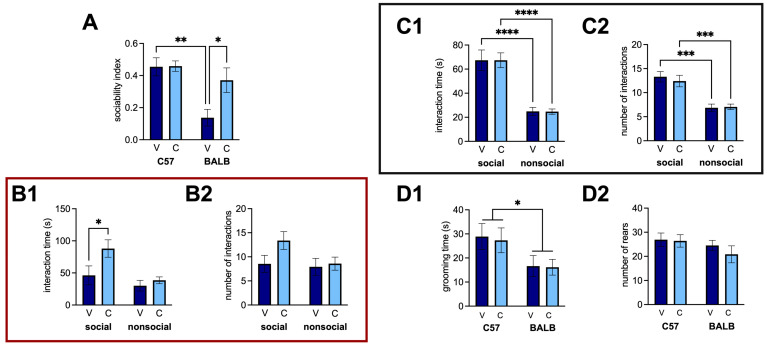
CEPO administration in the BALB mice recovered social approach deficits in the three-chamber task. (**A**) Sociability index in the three-chamber task for each strain. Red box denotes (**B1**) interaction time and (**B2**) number of interactions with each object for BALB mice. Dark gray box denotes (**C1**) Interaction time and (**C2**) number of interactions for C57 mice. (**D1**) Grooming time and (**D2**) number of rears during the three-chamber task for each strain. V, vehicle injections, C, CEPO injections. Error bars represent ± standard error of the mean. Two-way ANOVA with planned comparisons: * *p* < 0.05; ** *p* < 0.01; *** *p* < 0.001; **** *p* < 0.0001.

**Figure 4 neurosci-06-00025-f004:**
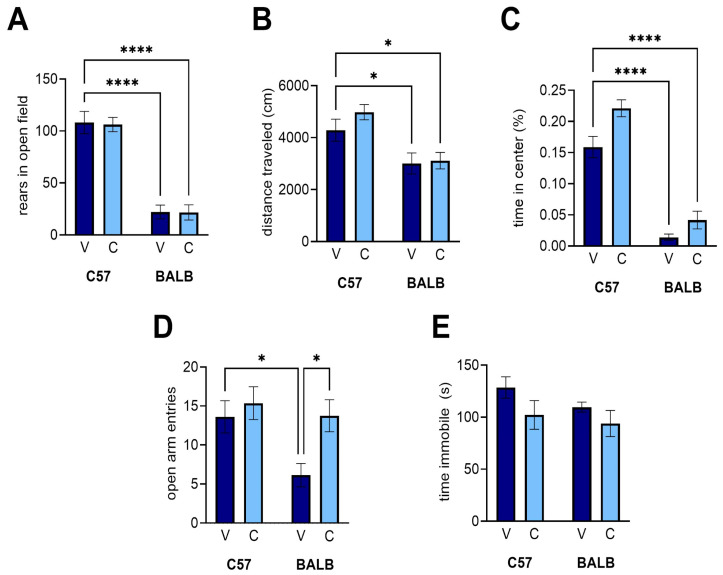
CEPO administration affected open arm entries in the elevated plus maze but not behavior in the open field task and the forced swim task. (**A**) Rears in the open field. (**B**) Distance traveled in the open field. (**C**) Percentage of time spent in the center of the open field. (**D**) Open arm entries, elevated plus maze. (**E**) Immobility time, forced swim task. V, vehicle injections, C, CEPO injections. Error bars represent ± standard error of the mean. Two-way ANOVA with planned comparisons: * *p* < 0.05; **** *p* < 0.0001.

**Table 1 neurosci-06-00025-t001:** Number of animals used in this study.

	BALB (*n* = 24 Total)	C57 (*n* = 21 Total)
Carbamoylated erythropoietin (CEPO, 40 μg/kg; *n* = 23 total)	12	11
Vehicle (*n* = 22 total)	12	10

## Data Availability

The original contributions presented in this study are included in the article and Appendix A. Further inquiries can be directed to the corresponding author.

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
