# Peer review of "Carbamoylated Erythropoietin Rescues Autism-Relevant Social Deficits in BALB/cJ Mice"

_neurosci, 2025, doi:10.3390/neurosci6010025_

Round 1

Reviewer 1 Report

Comments and Suggestions for Authors

The authors  did a great job, testing for social deficits in an autistic mouse models, using CEPO as the drug that potentially improves cognition and mood in rodents. The researches use 4 behavior studies to check if the administration of CEPO improves sociability in the mice models, which they indeed are suggesting via this publication. 

Some of the comments I have- 

Figure 2 A - using both mice BALB and C57The authors can modify the figure 2, using different colored outside boxes to show in A (2 mice models ) ,B1-B2 (BALB) , C1-C2 (C57)  and D1-D2 (c57 and BALB).  

I would have love to see the comparison of CEPO administration with 1-2 more analogs and how this analog is superior to others. 

Reviewer 2 Report

Comments and Suggestions for Authors

In this manuscript, Authors evaluated if Carbamoylated erythropoietin (CEPO), a chemically engineered non-hematopoietic derivative of EPO, can rescue behavioural phenotypes related to ASD. To reach this aim BALB/cJ (BALB) mice were used based on low sociability and anxiety-related behavioural deficits observed in this mouse model, as compared to the more social C57BL/6J controls. Mice were tested in four behavioural tasks: three-chamber, open field, elevated plus maze and Porsolt’s forced swim test, after the administration of CEPO (40 μg/kg), intraperitoneally every other day for 21 days at eight weeks of age. Results demonstrated that CEPO can rescue deficits in the three-chamber social approach task and the elevated plus maze task, but not in the open field or forced swim tasks in BALB/c mice, suggesting a potential therapeutic use of CEPO in ASD.

The paper is well written, the rational of the study is clear and the experimental methods and analysis are satisfactory. Some few questions:

Major points:

1. Line 84 and line 429: BALB/c is considered an idiopathic mouse model with face validity for ASD. I don’t agree with this concept. The face validity is not sufficient alone to label BALB/C mouse as an idiopathic model. It only implies that BALB/C mouse exhibits phenotypic features resembling autism in terms of behavioural profile but it is not necessary a model of ASD. Construct and predictive validity should be identified to establish a robust disease model. For this reason, the notion that BALB/C as idiopathic model of ASD in not correct and should be reconsidered, along with the title of the paper.

Minor points:

1. Legend fig 1: CEPO was administered in mice starting from postnatal week 8 as described in the text and in fig 1, but in the legend of Fig 1. “9 weeks postnatal” is reported.

2. Line 77: CEPO is safe for at least 10 doses, the references don’t support the sentence, CEPO treatment in these papers doesn’t exceed 4 administrations. In addition, Leconte et al., 2011 also demonstrated that CEPO dosed at 40 mg/kg twice per week over 45 days is safe.

3. The difference in the time spent grooming between strains of mice as reported in line 281 (p=0.0269) should also be shown in fig 3D1

Round 2

Reviewer 2 Report

Comments and Suggestions for Authors

All issues have been solved